# Molecular Dynamics Assessment of Mechanical Properties of the Thin Filaments in Cardiac Muscle

**DOI:** 10.3390/ijms24054792

**Published:** 2023-03-01

**Authors:** Natalia A. Koubassova, Andrey K. Tsaturyan

**Affiliations:** Institute of Mechanics, Lomonosov Moscow University, Moscow 119991, Russia

**Keywords:** actin, troponin, tropomyosin, thin filament, regulation of muscle contraction, molecular dynamics, stiffness

## Abstract

Contraction of cardiac muscle is regulated by Ca${}^{2+}$ ions via regulatory proteins, troponin (Tn), and tropomyosin (Tpm) associated with the thin (actin) filaments in myocardial sarcomeres. The binding of Ca${}^{2+}$ to a Tn subunit causes mechanical and structural changes in the multiprotein regulatory complex. Recent cryo-electron microscopy (cryo-EM) models of the complex allow one to study the dynamic and mechanical properties of the complex using molecular dynamics (MD). Here we describe two refined models of the thin filament in the calcium-free state that include protein fragments unresolved by cryo-EM and reconstructed using structure prediction software. The parameters of the actin helix and the bending, longitudinal, and torsional stiffness of the filaments estimated from the MD simulations performed with these models were close to those found experimentally. However, problems revealed from the MD simulation suggest that the models require further refinement by improving the protein–protein interaction in some regions of the complex. The use of relatively long refined models of the regulatory complex of the thin filament allows one to perform MD simulation of the molecular mechanism of Ca${}^{2+}$ regulation of contraction without additional constraints and study the effects of cardiomyopathy-associated mutation of the thin filament proteins of cardiac muscle.

## 1. Introduction

Contraction and relaxation of striated muscles are controlled by Ca${}^{2+}$ ions via the regulatory proteins tropomyosin (Tpm) and troponin (Tn) associated with the thin filaments. Tpm is a ca. 40 nm long parallel coiled-coil dimer of a helical shape. Tpm molecules bind each other via an overlap junction and polymerize to a long strand that binds F-actin. Two Tpm strands lie on both sides of the thin filament in helical grooves on the surface of the F-actin helix [1,2]. Tn is a complex of three protein subunits: Tn-C, Tn-I, and Tn-T, which binds Tpm with 1:1 stoichiometry [3]. Tn-C is a Ca${}^{2+}$ sensor that reversibly and specifically binds Ca${}^{2+}$ ions. Tn-I binds two other Tn subunits and Tpm. Its inhibitory domain anchors the Tn–Tpm complex on actin in a position that prevents the binding of myosin heads to actin. Tn-T also binds two other Tn subunits, Tpm, and F-actin and plays mainly a structural role, holding the whole regulatory complex together. The steric blocking model was put forward 50 years ago mainly based on structural data [4,5,6,7]. It explained Ca${}^{2+}$ regulation as follows [8]. In the absence of Ca${}^{2+}$, Tn holds the Tpm strand in a position where it blocks myosin binding sites on F-actin and keeps the muscle relaxed. Ca${}^{2+}$ binding to Tn causes conformational changes in the Tn–Tpm complex, leading to a rotation of the Tpm strand around the F-actin axis. The myosin-binding sites on actin become available for myosin, and the muscle contracts.

Later, McKillop and Geeves [9] developed a more detailed three-state model based on biochemical data. In this model, Ca${}^{2+}$ binding to Tn-C releases the Tpm–Tn complex from the ‘blocked’ state. However, the Ca${}^{2+}$ binding itself does not open the complex completely. In the ‘closed’ state, actin becomes available only for weak, non-stereo-specific binding of myosin heads. The subsequent transition of the weakly bound actin–myosin complex to a strongly and stereo-specially bound state leads to a further shift of the Tpm–Tn strand to the ‘open’ state. In this state, neighbor actin monomers also become available for myosin binding and the development of pulling force.

The main sources of structural information about thin filaments in different functional states are electron microscopy (EM) and X-ray diffraction [10,11]. The first model combining the data from both techniques had 13 actin monomers in 6 left-handed turns [12], with an axial repeat of 13 × 2.75 nm = 35.75 nm and a helical angle between the monomers of 167.1${}^{\circ }$. Growing structural data about Tpm, its position on actin, and the changes of this position upon activation led to a slightly different model that had 28/13 symmetry [13]. In this model, the F-actin period was equal to a longer ∼38.5 nm period of the Tpm–Tn complex, which was known to cover 7 × 5.5 nm consecutive actin monomers on each side of the filaments, providing a Tpm:actin stoichiometry of 1:7. The F-actin pitch was shown to change by ∼1% upon mechanical stretch [14,15], binding of myosin heads [16], or Ca${}^{2+}$ binding to Tn-C that releases Tpm from the blocked state [17]. Importantly, X-ray diffraction data suggest the F-actin pitch of 36.0–37.2 nm in live muscle [17], somewhat longer than 13 actin monomers (∼35.75 nm) and significantly shorter than 14 monomers (∼38.5 nm) equal to the Tpm–Tn period.

The structures of the core Tn complex formed by parts of all three Tn subunits were solved in the presence and absence of Ca${}^{2+}$ [18]. Ca${}^{2+}$ binding to Tn-C opens a hydrophobic pocket between the Ca${}^{2+}$-binding loop and the central α-helix of Tn-C. The switch domain of Tn-I forms a short α-helix that binds the pocket. Upon this binding, the C-terminal of Tn-I presumably undergoes a conformational change so that its inhibitory domain cannot bind actin. The unbinding of Tn-I from actin and Tpm releases the Tpm–Tn strand from the blocked state.

Recently, cryo-electron microscopy (cryo-EM) allowed the visualization of structural details in the thin filament in all states with near-atomic resolution (4.8–6.6 Å) [19,20,21,22,23,24]. The cryo-EM models revealed many important details of the regulatory complex. In the blocked state, the C-terminal part of Tn-I stretches along the Tpm strand from the central Tn-core-binding part of Tpm towards its N-terminal. The very C-terminal part of Tn-I kinks to form an anchor on an actin monomer that holds the strand in the blocked state. The N-terminal region of Tn-T binds the Tpm overlap region on a Tpm strand while its C-terminal region is a part of the Tn core on another strand. As the strands are shifted along the filament axis by ∼2.75 nm, the axial shift between the N- and C-terminal parts of Tn-T molecules bound to different Tpm strands on the opposite sides of a thin filament differ by ∼5.5 nm. The links between the N- and C-terminal parts of Tn-T were not resolved in the cryo-EM electron density maps. On the other hand, these links presumably provide a cooperative interaction between the two strands and cause a displacement of Tpm upon Ca${}^{2+}$ binding to Tn-C on the opposite side of the filament [21,22]. Despite the absence of the important part of Tn-T in the original model structures [20], the axial length of Tpm in these models is between 36 nm and 36.6 nm, which is much shorter than the ∼38.5 nm measured with low-angle X-ray diffraction.

Several computational approaches including fine positioning of oppositely charged actin and Tpm residues and correction of the α-helical pitch of Tpm in the original Yamada et al. [20] structures lead to the significantly refined models of the actin–Tpm–Tn complexes, 7uti and 7utl, in the presence and the absence of Ca${}^{2+}$, respectively [25] (see the reviews [3,26] for more details). Recently, Deranek et al. [27] proposed a model with refined central parts of Tn-T obtained with molecular modelling and structure prediction software. The model was verified by comparison with FRET measurements with fluorescent labels on different Tn-T residues and Cys374 on actin. These experimental and theoretical breakthroughs significantly improved our understanding of the structural basis of Ca${}^{2+}$ regulation of cardiac muscle contraction. Moreover, the near-atomic models of the cardiac thin filaments provided a powerful tool for understanding the mechanism of impairing cardiac function upon changes in genes encoding thin filament proteins. Along with variants associated with genetic cardiomyopathies, there are hundreds of genetic variants of unknown significance in the ClinVar database (https://www.ncbi.nlm.nih.gov/clinvar/ (accessed on 14 February 2023)) for Tpm, Tn-C, Tn-I, Tn-T, and actin. Knowing the intra- and intermolecular interactions for each amino acid may help in understanding the mechanism of functional changes caused by its substitution for another one, deletion or insertion and even finding a treatment for the elimination of these changes [3,28,29,30]. It is worth mentioning that mutations in cardiac Tn-T encoded with the TNNT2 gene more often cause genetic cardiomyopathies than mutations in any other thin filament proteins [31]. In the ClinVar database, there are tens of genetic variants which involve a majority of the Tn-T residues 150–200 not resolved in the structures of the regulatory units of cardiac thin filaments. Some of these mutations were characterized as pathogenic, while some are of uncertain clinical significance.

The Ca${}^{2+}$ regulation of cardiac muscle contraction is a highly dynamic process, and dynamic and mechanical properties of the regulatory system are crucially important for its functioning. Atomistic molecular dynamics (MD) simulation along with experimental approaches are the tools for studying these properties. Early MD studies of actin filaments using low-resolution structures of F-actin available at that time revealed the effect of MgADP or MgATP and the DNA-binding loop on the mechanical and structural characteristics of the filaments [32,33,34,35,36]. From the MD trajectories, axial and angular periods of the F-actin helix and its longitudinal, torsional, and bending stiffness were estimated [32,33,34,35,36]. The values obtained from MD simulations were within the range of their experimental estimates. We [37] performed 204.8 ns long MD simulations of a 24-monomer F-actin fragment using the modern high-resolution structure and different force fields in the presence and absence of MgADP. It was found the reliable estimates of axial and angular parameters of the F-actin helix can be obtained from the MD trajectories provided that two actin monomers on both ends of the segment were excluded from the analysis.

MD simulation of the F-actin–Tpm complexes [38,39,40] and more recent works which used high-resolution structures of the regulatory system [25,41,42,43,44] revealed many important characteristics of the system, including the effects of genetic variants and posttranslational modifications of actin, Tpm, and Tn. These works also revealed some problems with the MD simulation of the thin filaments. Even in a relatively long model, the ends of Tpm–Tn strands tend to move away from F-actin if no additional constraints are applied. The constraints themselves may introduce artificial forces that may affect the system. For example, Pavadai et al. [42] used a model consisting of 28 actin monomers and 4 Tpm–Tn complexes assuming that the F-actin and Tpm–Tn periods are equal to 38.5 nm and used periodic boundary conditions. As explained above, the X-ray data suggest that the F-actin period is shorter than 14 monomers, so the periodic boundary conditions possibly introduced some torque that may compromise the results of the simulations. Here, we present the results of MD simulations using new atomic models of 26-monomer-long fragments of the F-actin–Tpm or F-actin–Tpm–Tn filaments in explicit solvent without any additional constraints. In the models of the fully reconstructed F-actin–Tpm–Tn filaments, the central parts of Tn-T molecules which connect two Tpm–Tn strands were built using the structure prediction software. The results are compared with available experimental data and discussed in terms of the quality of currently developed models for a meaningful and reliable description of the mechanical properties of thin filaments in cardiac muscle.

## 2. Results

### 2.1. Structural Model of the Thin Filament in Ca-Free State, Actin Helix Parameters

The previous MD studies of thin filaments showed that the models became unstable unless tropomyosin ends and/or actin monomers were subjected to restraints imitating the effect of a continuous tropomyosin cable [45,46]. Our aim was a model long enough to place four Tn complexes and be stable in prolonged MD runs without additional constraints. We started with the available 6kn7 PDB data of the regulatory complex in the blocked state [20]. The actin monomers in this structure do not follow pure helical symmetry: after the alignment of the filament axis with the *z*-axis, the average axial distance between the monomers, *d*, and the average helical angle between adjacent monomers, φ, estimated from the positions of the centers of mass of the monomers were: d=2.75±0.04 nm, $\varphi =166.6\;\pm \;6.6^{\circ }$ (M±SD,n = 14 neighboring pairs). These helix parameters correspond to an average pitch of the actin helix of 37.1 nm, close to that measured with X-ray diffraction in live muscle [17]. The tropomyosin α-helix in this model had a residue rise of 0.142 nm [42], which was shorter than ∼0.15 nm in Tpm structures determined with protein crystallography [47,48,49]. Furthermore, the axial distance between the residues with the same numbers on the neighbor Tpm molecules in the same strand varied from 36 nm to 36.6 nm, i.e., much shorter than the axial period of Tn complexes (∼38.5 nm) [50].

The refined Tpm with an increased residue rise and corrected position on actin in the blocked state was suggested [51]. The reported structure included 8 actin monomers and the central parts of Tpm molecules (residues 50–220) together with residues 137–210 of Tn-I. Our first model is shown in Figure 1. The details of model construction are given in Methods. It is 26 actin monomers long and includes two copies of the ’refined’ fragment [51] placed at an axial distance of 38.5 nm between the centers to ensure the correct Tpm length. The missing inner and outer parts were taken from 6kn7. The Tpm helices from 6kn7 were slightly extended along their skeleton axis to match ’refined’ Tpm fragments using homemade software based on the TWISTER algorithm [52]. The central parts of all Tn-T molecules, residues 151–198 (here and after the residues are numbered as in 6kn7), were built using AlphaFold and Modeller software as described in Methods to obtain the whole Tpm–Tn regulatory complex except the first 86 N-terminal residues of Tn-T not resolved in any cryo-EM electron density map. Two Tn complexes (and Tpm–Tn, respectively) on the opposite sides of the thin filament are not equivalent. Following [3], we denote the Tn complex that lies closer to the pointed F-actin end than its neighbor on the opposite side of the filament as Tn1 (its Tn-T subunit is shown hot pink in Figure 1) and another Tn complex as Tn2 (Tn-T shown magenta in Figure 1). Tpm strands 1 and 2 are those which are bound to the core of Tn1 and Tn2, respectively. We also denote actin long pseudo-helices which contact Tpm–Tn strands 1 and 2 as 1st and 2nd (light gray and dark gray in Figure 1, respectively). A 102.4 ns MD run was performed on the 1st model.

In 2022, a new 7utl structure of thin filament in the blocked state was released [26]. F-actin in this model was also not quite helical, and the average helix parameters differ from those of 6kn7: d=2.72±0.07 nm, $\varphi =167.3\pm 6.5^{\circ }$ (M±SD,n = 17 neighboring pairs). These parameters correspond to the helical pitch of 38.5 nm, the same as the axial period of the Tpm–Tn complex found in the X-ray diffraction experiments [16,17]. Our second model was built from the 2 copies of the 7utl atomic model, shifted and rotated with respect to the actin filament axis so that tropomyosin cables match at their meeting points (Methods) and also included 26 actin monomers, 4 Tn complexes, and Tpm strands twisting around an actin filament fragment. Models 1 and 2 fitted the electron density map of the thin filament in the blocked state [20] and were similar to each other, although the pitches of their long F-actin helices and the helical rise of Tpm α-helices differ slightly, as discussed above. A minor difference was that one of the Tpm α-helices of one of two Tpm–Tn strands in the region of the 2nd pseudo-repeat (P2, Tpm residues 50–70) near an edge of the electron density map was shifted away from the F-actin axis by a distance of up to 0.66 nm. Model 2 is shown in Appendix A. Several short MD runs were performed on this model, with the longest one being 43.2 ns. In all runs, 3 of 4 Tn-C subunits flew away and detached from Tpm. Thus, after repeating a few shorter efforts, a 204.8 ns long run was performed with the F-actin–Tpm complex without Tn. The snapshots of the MD trajectories were taken every 200 ps and used for further analysis.

Note that Tn-T bridges between two Tpm–Tn strands are different as the core parts of the Tn1 and Tn2 complexes are shifted axially by ∼2.75 nm and the axial distances between the N- and C-termini of these Tn-T molecules 1 and 2 differ by ∼5.5 nm (Figure 1). The actin helix parameters, *d* and φ, were calculated for each time frame. The apparent helical period *P* of F-actin was calculated as
(1)$$P=d\frac{\pi }{\pi -\varphi }.$$

The fluctuations of *d*, φ, and the radial distance *r* of the axis of the Tpm-coiled coil from the *z*-axis were higher at the ends of the model, so a pair of monomers from both sides was excluded from the further analysis. The results of the statistical analysis of the helix parameters for three MD trajectories are shown in Table 1.

The parameters of the F-actin helix, *d* and *P*, converged to values slightly lower than those of the original Models 1 and 2 and were found experimentally with low-angle X-ray diffraction.

### 2.2. Mechanical Properties of the Filaments

Mechanical parameters, including the persistence length ξ, bending $K_{b}$, axial *K*, and torsional *G* stiffness, were calculated as described [37] from the MD trajectories (see also Methods) and are presented in Table 2. The values were close and somewhat higher than those estimated during an MD simulation of F-actin [37].

For the short MD run of Model 2 with Tn complexes, the torsional stiffness was close to that previously found for F-actin [37], while for Model 1 and the long MD run of Model 2 without Tn (especially its second half), the torsional stiffness was significantly higher (Table 2). To dissect an additional rotational degree of freedom of F-actin that affects its interaction with Tpm during the three MD runs, we analyzed angular fluctuations of long actin helices 1 and 2 (light gray and dark gray in Figure 1) as described in Methods. The results are shown in Figure 2.

Except for high amplitude fluctuation on the ends of the structure, ∼20${}^{\circ }$ fluctuations in the middle of the filament were observed for the long actin helix 2 during the short MD run of Model 2 with Tn complexes (Figure 2). This observation shows substantial mobility of the long actin helices around the F-actin axis.

### 2.3. Fluctuations of the Angular and Radial Position of Tpm on F-Actin

The time-averaged angular fluctuations relative to the F-actin axes for the three MD runs are shown in the Appendix A. As actin monomers themselves fluctuated during the MD simulations, a more useful representation of Tpm fluctuation is relative to the long F-actin helices or, more correctly, to the long pseudo-helices formed by actin monomers regulated by each Tpm strand. The long helices were approximated by the linear regression lines on the (azimuthal angle, axial distance) plane for Lys328 residues of 13 actin monomers in each helix. The fluctuations of the azimuthal differences between the position of the Tpm residues and the time averaged root-mean-squared differences are shown in Figure 3 against the distance along the F-actin axis. The same data plotted against the Tpm residue number is shown in Appendix A. Radial fluctuations of Tpm with respect to the F-actin axes for 3 MD runs are shown in Figure 4. The same fluctuations plotted against Tpm residue numbers are shown in Appendix A.

Apart from highly mobile Tpm ends caused by artificial cuts of long Tpm–Tn strands, the root-mean-squared angular Tpm fluctuation with respect to neighbor long actin helices was less than 14${}^{\circ }$. For a short MD run of Model 2 with Tn, the amplitude of the fluctuations was somewhat higher than for the two other MD runs.

The angular and radial Tpm fluctuations near the edges of the model structures were more intense than those in their central parts for all 3 runs, probably due to the absence of constraints, despite our effort to improve the actin–Tpm contact by adding a second overlap zone in Model 2. For Model 2, there was also high-amplitude angular and radial Tpm fluctuation in the middle of the structure. The radial fluctuation with SD of 0.5–0.6 nm suggests a loss of contacts of the Tpm strand with 2 or 3 actin monomers for a substantial fraction of the MD trajectory (Figure 4). Surprisingly, the removal of the Tn complex from Model 2 only marginally affected the fluctuations and even suppressed the radial ones, although the N-terminal part of Tn-T presumably stabilizes the junction [26].

The high-amplitude fluctuations in the middle parts of the Tpm strands (Figure 3 and Figure 4) were not accompanied by a significant increase in the time-average radial distance of this part of Tpm from the filament axis, while at the end of the structure, some Tpm molecules moved away from actin Appendix A.

To find out how the actin–Tpm interaction was affected by these fluctuations, we estimated the relative lifetime of hydrogen bonds (h-bonds) between the Lys326 and Lys328 residues of actin and Tpm residues. In each time frame, the h-bond occupancy was set to 1 if at least one h-bond existed; otherwise, it was 0. Values averaged over time are shown in Figure 5. The h-bonding and electrostatic interaction are believed to be involved in the stabilization of the blocked state of the thin filament [53].

A low lifetime of h-bonds between Lys326 and Lys328 of an actin monomer and Tpm strand 2 was observed during a short (43 ns long) MD run of Model 2 in the presence of Tn complexes. It was <5%, suggesting a local break of the blocked state of the regulatory system (Figure 5). For Model 1, the lifetime was about 90%, except for the very ends of the structure where Tpm strands were cut.

For the 204.8 ns long MD run of Model 2 without Tn, a decrease in the h-bond lifetimes in the middle of the stucture was also observed. On average, it was less severe than that for a shorter run in the presence of Tn in Figure 5. The difference was not probably caused by the absence of Tn but rather by the time of the MD trajectory, as during the long MD trajectory, the h-bond lifetime gradually improved: during the first 51.2 ns of the MD trajectory, it was below 40% for both Tpm strands and became 60–75% for the last quarter of the MD trajectory Appendix A.

## 3. Discussion

The results suggest that a sufficiently long atomic model of the thin filament in which a Tpm strand wraps around F-actin by ca. 360${}^{\circ }$ can be used for MD simulations without additional constraints. Although Tpm ends unravel slightly and move away from actin, the rest of the Tpm strand remained in the vicinity of its surface (Appendix A) and fluctuated azimuthally within a range of ca. 10${}^{\circ }$ (Figure 3) and in general remained in close contact with actin (Figure 5). However, local high-amplitude azimuthal (Figure 3) and radial (Figure 4) fluctuations of Tpm in the middle of the structure and a break of the h-bonding of Tpm and an actin monomer (Figure 5) observed in the short MD simulation with Model 2 may indicate the break of Gestalt complementarity of Tpm and F-actin [54] or poor quality of the inter-protein interface, especially in some region of Model 2.

The azimuthal fluctuations of the central part of the Tpm strand for Model 1 (Figure 3) were within the range estimated from the analysis and modeling of X-ray diffraction data [55]. Such fluctuations were small enough to keep the blocked state of the thin filament in the absence of Ca${}^{2+}$.

Two models of the thin filaments with different starting F-actin structures converged to a structure with helical parameters *d* and φ (Table 1), close to those estimated from MD simulations of F-actin alone [37] and those found experimentally with low-angle X-ray diffraction [17]. Model 1 predicted a slightly longer axial helical step *d* than Model 2, although both values were within their experimental range. The long actin helical pitch *P* obtained from the MD simulation was slightly shorter than found experimentally and significantly shorter than the axial distance between the consecutive Tn complexes of ∼38.5 nm. The average *P* value in the MD runs of Model 2 was significantly shorter than its initial value, suggesting possible intrinsic torque inherited from the atomic model.

Surprisingly, the removal of Tn from Model 2 caused an elongation of the long F-actin pitch, *P* (Table 1), while in experiments with live frog muscle, *P* shortened slightly upon activation and release of Tn-I from actin [17].

The longitudinal, torsional, and bending stiffness characteristics of the thin filament for both models were similar, although somewhat higher than those estimated for F-actin alone [37]. However, the effect of Tpm with or without Tn on the bending stiffness and persistence length was somewhat less pronounced than was found in EM experiments [56], although the absolute values were quite close. Our MD estimates of the longitudinal and torsional stiffness of the thin filaments were also within the range of their experimental values reviewed and discussed by Deriu et al. [35].

The interaction of conserved Tpm negatively charged residues with actin [53] was supported during the MD trajectories, except for the very ends and several actin monomers in the middle of the structure in Model 2 where Tpm–actin h-bonds were impaired (Figure 5). The reason for the local destabilization of the F-actin–Tpm interaction was probably the poor quality of Model 2 in the docking area of two 7utl structures. This explanation is supported by the observation that the Tpm–actin h-bonds gradually improved during the long MD simulation of the same Model 2 without Tn (Appendix A), suggesting that further refinement of Model 2 may improve its quality.

Another surprising and disappointing result of our MD simulations was the instability of the Tpm–Tn-C contact for both models. During long MD trajectories, Tn-C and a part of the Tn core domain that holds it on the surface of the F-actin–Tpm filament often flew away from Tpm. This was, probably, caused by the poor quality of the Tpm–Tn-C contact in both models. The resolution of the cryo-EM electron density maps [20,21] is good enough for positioning the N- and C-terminal lobes of Tn-C to the Tpm-coiled coil. However, this was insufficient to determine the amino acids involved in the contact between these proteins. A possible approach for solving this problem is a structure refinement, i.e., improvement of the Tpm contact with Tn-C and, possibly, Tn-I and Tn-T domains of the Tn core complex using available software. Other regions of the models that require further refinement are the Tpm overlap junction and its contact with the N-terminal part of Tn-T, the docking region between two copies of the 7utl models where the resolution of the electron density maps is reduced, and central parts of the Tn-T molecules built with structure prediction software. The refinement can be done by carefully minimizing the energy of the problem regions of the models while keeping the remainder of the model fixed or heavily constrained.

## 4. Materials and Methods

The X-ray diffraction data show that the axial periods of the actin helix and the Tpm–Tn complex do not coincide. Both our models of the regulatory units had 26 actin monomers, close to the doubled period of the actin helix, and the axial Tpm length was close to the full period of the Tpm helix, placing four Tn complexes.

The first model (Model 1) of the long regulatory unit was built using the 6kn7 structure [20] and the ‘refined’ structure of an 8-actin-monomer-long fragment of the thin filament in the blocked state with fragments of 2 Tpm molecules from both strands and the C-terminal fragment of Tn-I [51]. Both structures were aligned so that the F-actin axis coincided with the *z*-axis, and cylindrical coordinates of the centers of mass of actin monomers were used to describe their positions. The ‘refined’ structure was fitted to 6kn7 using the least square fit with UCSF CHIMERA [57] by all 8 × 375 = 3000 C${}_{\alpha }$ atoms of 8 actin monomers. As both structures were fit to the same cryo-EM density map [20], the translation was negligibly small, and the ‘B-I’ chains of actin monomers in the original 6kn7 structure were replaced with the ‘refined’ actin monomers.

The central parts of the Tpm molecules (residues 50–220) were also taken from the ‘refined’ structure where the Tpm structures follow the canonical α-helical symmetry. The ‘refined’ and ‘original’ Tpms from 6kn7 structure do not match: 6kn7 Tpm had a shorter helical rise [42]. We used homemade software to bring them together. The overlap junction between neighbor Tpm molecules (residues 280–284) remained at the same positions as in the 6kn7 structure. The skeletons of the Tpm α-helices were calculated using the TWISTER algorithm [52], and the 2nd halves of the Tpm dimers of the 6kn7 were slightly extended along their skeletons from the 280th residue towards the center. The extension coefficient was varied until the positions of C${}_{\alpha }$ atoms of at least one residue from the extended and ’refined’ Tpm helix in the dimer coincided with a precision of 0.5 Å or better. Both dimers were extended by a factor of 1.082 with matching points at the 189th and 200th residues in the 1st and the 2nd strand, respectively.

The 12-monomer-long fragment ‘B-M’ that included 8 ‘refined’ and 4 ‘original’ actin monomers together with 4 reconstructed 50–284 Tpm chains was copied and translated along the filament axis after the last 15th ‘O’ monomer of 6kn7 so that the characteristics of the actin helix, including the axial and azimuthal differences between the centers of mass of the neighbor actin monomers, were the same as their average values in the 6kn7 structure. The first ‘A’ monomer of 6kn7 was removed to limit the model length by 26 actin monomers. The gaps between the end of the overlap region in 6kn7 and the ‘refined’ Tpms in the translated half of the structure, residues 30–50, were too short to be filled even by a corresponding segment from the 6kn7 Tpm chains. The iterative search was performed in UCSF Chimera to find the ends of the (i, j) interval providing minimal RMSD for 3 C${}_{\alpha }$ pairs at both ends: for the residues i−1, i, i+1, and j−1, j, and j+1, correspondingly. The best match was achieved for (20,85) in the 1st strand and the (20,70) interval in the 2nd strand. The long Tpm strands in Model 1 had 2 molecules each: one truncated, residues 50–284 (red-gold in the 1st strand and purple-azure in the 2nd strand in Figure 1) and one full (blue-green in the 1st strand and brown-orange in the 2nd strand in Figure 1). All 4 Tpm dimers in the model were minimized separately in Gromacs [58] in two steps as described below and returned to the model matched by their C${}_{\alpha }$ atoms using UCSF Chimera. The complex of 26 actin monomers with Tpm molecules were again minimized.

Residues 151–198 of both Tn-T molecules are missing in the available to-date atomic models. To include them in our model, residues 151–180 were taken from the AF-P45379-F1 dataset for cardiac muscle Tn-T from the AlphaFold [59] predicted structure database (https://alphafold.ebi.ac.uk/ (accessed on 14 February 2023)). The α-helical structure in the 151–177 region in the predicted structure is considered as ‘confident’ according to a per-residue confidence score. The predicted structure was matched to Tn-T N-terminal fragments in 6kn7 by C${}_{\alpha }$ atoms of the 139–146 residues. Then, Modeller software [60] was used to generate a short missing loop (181–198). The top 7 most probable structures generated by Modeller were considered. As some of the predicted structures had clashes with actin, the non-clashing structure with the highest probability score was chosen and used for further minimization and MD modeling.

The second structural model (Model 2) of the regulatory unit of thin filaments was based on the 7utl atomic structure of human cardiac thin filament in the calcium-free state released in 2022 [26]. The 7utl structure includes 18 actin monomers, 4 tropomyosin coiled-coil strands around them, and 2 not completely resolved troponin complexes. The missing 151–198 residues of Tn-T subunits were reconstructed using the same protocol as for Model 1 with AlphaFold and Modeller, and the whole structure was subjected to energy minimization in Gromacs. Then a copy of the minimized structure shifted and rotated around the actin filament axis to produce the best match between the C${}_{\alpha }$ atoms from the four first actin monomers of the copy and the four last actin monomers of the origin structure. The shift and rotation also nearly superimposed 3 of 4 corresponding Tpm helices at the residues 50–65. The pairs with minimal distance between C${}_{\alpha }$ atoms in the copies were chosen as a connection points: residues 63, 53, and 51, respectively. One (locally outer, in the 1st strand) helix appeared slightly different in two copies, as residues 31–51 of the original structure were slightly compressed by a factor of 0.95 to meet the C${}_{\alpha }$ atoms of the 51st residues using homemade software. The central 26-actin-monomer-long fragment of the expanded structure was used for the calculations; 26 was the minimal number of actin monomers that place all 4 Tn complexes including Tn-T (residues 89–151; 199–272) and Tn-I (residues 41–210) parts resolved in the original structure.

The MD calculations performed using Model 1 showed large fluctuations of Tpm free ends. In Model 2, the first Tpm molecule was shortened to remove the ‘free’ residues not contacting actin monomers and included residues 67–284. The second molecule in the strand again had all 284 residues and from the ‘barbed’ end of the actin filament in was connected with the jointed residues 1–29 of the third molecule, forming an overlap.

### 4.1. MD Simulation

MD simulations were carried out in the Gromacs 2021.5 package [58]. The model was placed in a rectangular cuboid box, filled with water molecules; model TIP3P and the CHARMM36 force field [61] were used. The cell size was chosen so that the distance from the protein atoms to its boundaries was at least 1.5 nm, and periodic conditions were set at the borders. Na${}^{+}$ and Cl${}^{-}$ ions were added to ensure the electrical neutrality of the system and the ionic strength of 0.15 M. Energy minimization was carried out using the steepest descent method in two steps. The first step was carried out with harmonic constraints applied to the atoms of the protein backbone (N, C${}_{\alpha }$ and C) until a maximum force of 1000 kJ mol${}^{-1}$ nm${}^{-1}$ was reached. In the second stage, the restrictions were removed, and the energy was minimized until the maximum force became less than 300 kJ mol${}^{-1}$ nm${}^{-1}$. Then, the system was balanced in the NVT and NPT ensemble to a temperature of 300 K and a pressure of 1 atm using a Berendsen thermostat [62]. The duration of the MD trajectories varied from 43 ns to 204.8 ns with a step of 2 fs. Data were recorded with a step of 200 ps.

### 4.2. Analysis of MD Trajectories

All the analysis was made after removing the rigid body movements in all time frames as described in [37]. The center of mass of each of the 26 actin monomers was calculated. The least squares method was used to determine the filament axis—the straight line, the sum of the squared distances from which to the centers of mass of monomers is minimal. Then, the whole system was subjected to the orthogonal transformation that takes the unit vector of this line to the vector (0, 0, 1), so that the *z*-coordinate of the center of mass of the 26 centers of mass of actin monomers was at the origin and rotated azimuthally so that the linear regression line for φ(z) for all centers of mass of actin monomers had an intercept equal to zero.

The bending of the thin filament axis relative to the *z*-axis was determined as follows. In each time frame, the centers of mass of the actin monomers were projected to (x,z) and (y,z) planes and fitted with a parabola: (2)$$x=\frac{1}{2}k_{x}z^{2}+b_{x},\;y=\frac{1}{2}k_{y}z^{2}+b_{y}.$$

The cosine of the angle between the *z*-axis and the tangent to the parabola at the end of the actin filament segment with a half-length *l* was determined by the formula
(3)$$\cos\left(\alpha \right)=\sqrt{\frac{1}{1+l^{2}\left({\left(k_{x}\right)}^{2}+{\left(k_{y}\right)}^{2}\right)}}.$$

The persistence length, ξ, and bending stiffness, $K_{b}$, were estimated using worm-like chain theory [63]: (4)$$\langle \cos\left(\alpha \right)\rangle =\exp\left(\frac{-l}{\xi }\right),\;\xi =\frac{K_{b}}{k_{B}T},$$
where <...> denotes the time averaging over a total number of time frames, $k_{B}$ is the Boltzmann constant, and *T* is the absolute temperature.

The parameters defining the actin helix were calculated in each time frame. The average distance *d* between the centers of mass of adjacent monomers along the filament axis and the average helical angle between adjacent monomers φ. The longitudinal (*K*) and torsional (*G*) stiffness of the thin filament were evaluated from their fluctuations in the middle part of the filament to reduce errors caused by higher fluctuations of the fragment ends. We used the segment between the centers of mass of 5th and 6th monomers from both ends and measured *L*—its axial projection and Ψ—the difference between the polar angles.
(5)$$K\frac{{\langle \Delta L^{2}\rangle }_{\Delta t}}{2\langle L\rangle }=\frac{k_{B}T}{2},\;G\frac{{\langle \Delta \Psi^{2}\rangle }_{\Delta t}}{2\langle L\rangle }=\frac{k_{B}T}{2}.$$

Here, as before, the angle brackets denote time averaging over the trajectory, Δ is the deviation of the value from its average value, and Δt is the width of the sliding time window in which the variance of the corresponding value was determined. Here we used Δt = 16 ms, as it was found to be good enough in the actin filament MD analysis [37]. The averaging of the length dispersion *L* and the difference of azimuthal angles Ψ was carried out for the entire MD trajectory and separately for its second half in order to isolate the possible effects of the initial conditions and the establishment of a stationary regime of thermal fluctuations.

The position of Tpm strands was analyzed using their C${}_{\alpha }$ atoms in cylindrical coordinates relative to the actin filament axis and also relative to the corresponding actin long helix (light gray and dark gray in Figure 1). Long actin helices 1 and 2 wrapped by Tpm strands 1 and 2 were determined by the positions of the C${}_{\alpha }$ atoms of Lys328 actin residues. After calculations of their cylindrical (*r*, φ, *z*) coordinates and keeping φ-coordinate monotone over *z*, a linear regression line for φ(z) was calculated in each time frame for both long actin helices. Then, the difference between the angular actual position and the regression line was calculated for each C${}_{\alpha }$ atom of the Tpm residue. The differences were used to estimate azimuthal fluctuations of Tpm residues in the MD runs.

Hydrogen bonds were calculated using the hbond function available in Gromacs.

## 5. Conclusions

The use of relatively long atomic models of the thin filament of cardiac muscle that include F-actin, Tpm, and all components of Tn, including the bridges between two Tpm–Tn strands, allows one to perform MD simulations without additional constraints on the ends of the model structure. MD simulations predict the parameters of the F-actin helix and values of the bending, longitudinal, and torsional stiffness of the filaments close to those found experimentally. However, the models need further refinement to improve the Tpm–Tn contact and stability of the regulatory protein strands.

## Figures and Tables

**Figure 1 ijms-24-04792-f001:**
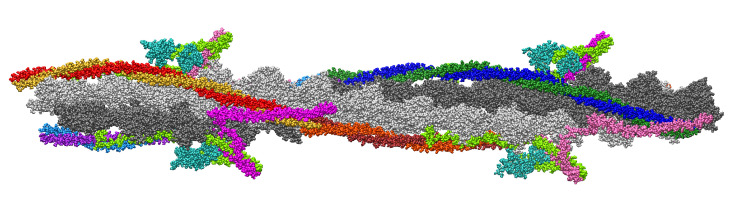
A refined atomic model (Model 1) of a segment of a thin filament of cardiac muscle. The pointed end of F-actin is on the left. The *z*-axis in the model coincides with F-actin axis and it is directed from left to right. Actin is shown in light gray and dark gray (long helices 1 and 2, respectively), strand 1 of Tpm is red and light brown or dark brown and orange; strand 2 is light blue and purple or blue and dark green; Tn-C is sea green, Tn-I is light green; two structurally different Tn-T are hot pink (Tn-T1) and magenta (Tn-T2).

**Figure 2 ijms-24-04792-f002:**
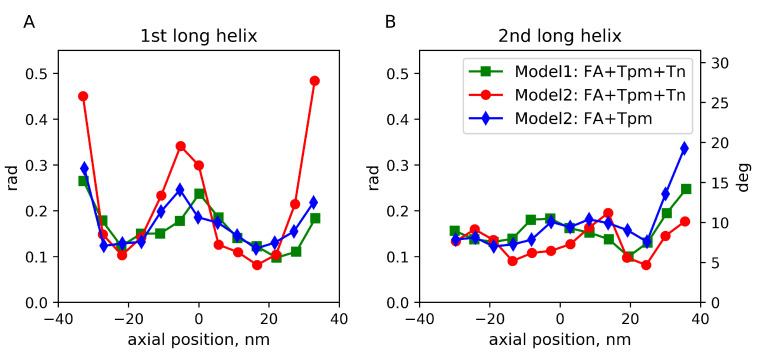
Root-mean-squared angular fluctuations of the long actin helices 1 (**A**) and 2 (**B**) bearing Tpm–Tn strands 1 and 2, respectively, for three MD runs: Model 1 (green) and Model 2 with Tn, short MD run (red) and without Tn, long MD run (blue).

**Figure 3 ijms-24-04792-f003:**
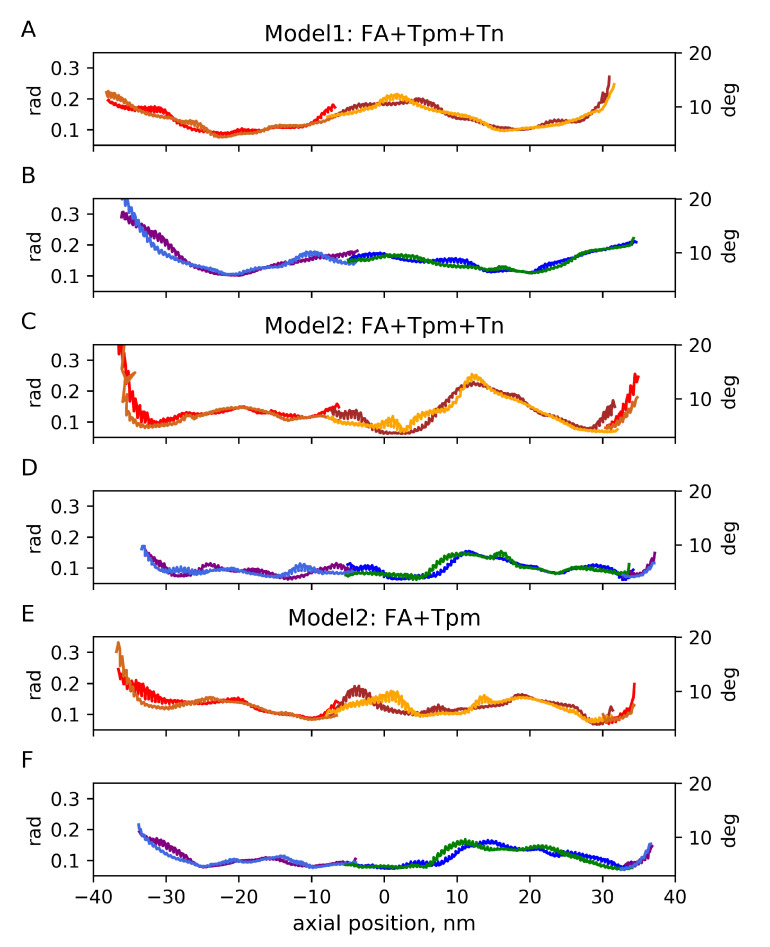
Time-averaged angular fluctuations (standard deviations, SD) of Tpm strands 1 (red/light brown and brown/orange; (**A**, **C**, and **E**) panels) and 2 (purple/light blue and blue/green; (**B**, **D**, and **F**) panels) with respect to long actin helices 1 and 2, respectively.

**Figure 4 ijms-24-04792-f004:**
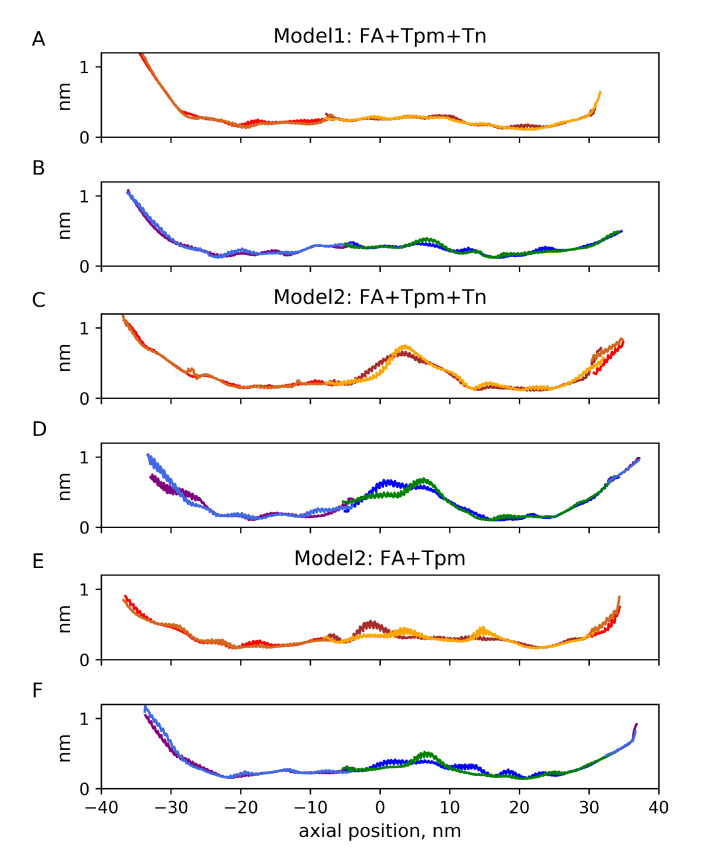
Time-averaged radial fluctuations of Tpm strands 1 (red/light brown and brown/orange; (**A**, **C**, and **E**) panels) and 2 (purple/light blue and blue/green; (**B**, **D**, and **F**) panels). The same color code as in Figure 3.

**Figure 5 ijms-24-04792-f005:**
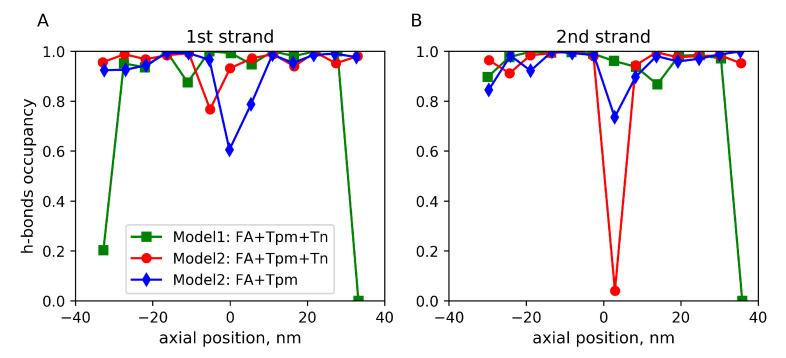
The relative lifetime of h-bonds between actin residues Lys326 or Lys328 of the long actin helices 1 (**A**) and 2 (**B**) and conserved acid Tpm residues of the corresponding strands 1 or 2 shown along the filament axis. Color code is the same as in Figure 2.

**Table 1 ijms-24-04792-t001:** Parameters of the F-actin (FA) helix estimated from the MD simulation.

Model Number and Description	Trajectory Duration, ns	*d*, nm	$\varphi {,}^{\;\circ }$	*P*, nm
1: FA+Tpm+Tn	104.2	2.75 (2.75) ^1^	166.2 (166.1)	35.8 (35.6)
2: FA+Tpm+Tn	43.2	2.72	166.0	35.0
2: FA+Tpm	204.8	2.71 (2.71)	166.5 (166.2)	36.1 (35.5)

^1^ Values in brackets are calculated for the 2nd half of the long MD trajectories.

**Table 2 ijms-24-04792-t002:** The values of the persistence length, ξ, bending, $K_{b}$, longitudinal, *K*, and torsional, *G*, stiffness estimated from MD trajectories.

Model Number and Description	ξ, μm	$K_{b}$, 10${}^{-26}$ Nm${}^{2}$	*K*, 10${}^{-9}$ N	*G*, 10${}^{-26}$ N
1: FA+Tpm+Tn	6.580	2.7	48.3 (51.6) ^1^	3.96 (4.45)
2: FA+Tpm+Tn	8.370	3.5	42.1	2.88
2: FA+Tpm	8.061	3.3	41.0 (35.9)	5.87 (7.05)

^1^ Values in brackets are calculated for the 2nd half of the long MD trajectories.

## Data Availability

The data presented in this study are available on request from the corresponding author.

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
