# Peer review of "Molecular Dynamics Assessment of Mechanical Properties of the Thin Filaments in Cardiac Muscle"

_ijms, 2023, doi:10.3390/ijms24054792_

Round 1

Reviewer 1 Report

Dear Authors,

I congratulate all the Authors for their contributions to the writing of the manuscript entitled “Molecular dynamics assessment of mechanical properties of the thin filaments in cardiac muscle”.

I have few comments on the manuscript:

1.    Lines 19-29: Please include references where appropriate. Without references, these statements are highly speculative.

2.    Line 169: Where is the 2nd model? Side-by-side comparison to the model 1 is recommended. What are the structural differences between these two models?

3.    First part of the methodology is redundant and difficult to follow. The Authors should concisely describe the protocols used in the study so that the results could easily be reproduced.

I consider the manuscript is sufficiently comprehensive and can be considered for publication in the IJMS journal after these issues have been properly addressed.

My sincere congratulations to all Authors.

Author Response

We are very grateful for the evaluation of our work and very useful and helpful suggestions. We revised the MS according to the recommendations.

Lines 19-29: Please include references where appropriate. Without references, these statements are highly speculative.

We agree with the suggestion. References are added.

Line 169: Where is the 2nd model? Side-by-side comparison to the model 1 is recommended. What are the structural differences between these two models?

We added a comparison of Models 1 and 2 in the Results section of the revised MS and added a Supplementary Material Figure S1 with Model 2 that can be compared to Model 1 in the main text, Figure 1.

First part of the methodology is redundant and difficult to follow. The Authors should concisely describe the protocols used in the study so that the results could easily be reproduced.

We rewrote the first part of the Method section and added methodological details needed to reproduce the results as suggested.

Reviewer 2 Report

The manuscript by Koubassova and Tsaturyan describes molecular dynamics simulations performed on current thin filament models over long simulation time periods to determine the stability and quality of the published thin filament models. The simulations have been run over greater than 100 ns, which is long compared to other published studies in the field. They find that the tropomyosin model shows large variability in its radius from the actin axis, suggesting that the tropomyosin in the model is losing its contact with actin. The authors conclude that further refinement of the structures is needed to fix these errors in the models. The authors make an important point about the quality of the models we use to predict the mechanism of mutations that lead to cardiomyopathies, for example, and experiments like these are needed to avoid incorrect conclusions later on.

However, the data used to support the conclusions and the proposed source of the errors is incomplete. For instance, a simple explanation for the poor behavior of the tropomyosin model in the simulations would be the extension of the tropomyosin the authors used with their homemade programs to fill out missing parts of the model before simulation. Another potential source of error is the TnT midpiece structure the authors built. Neither of these were independently tested to see what their effect would be on the global structure. The effects on radial fluctuation reported could be a result of poor contacts in the initial structure from these additional sections which then propagate out into weak points in the tropomyosin coiled coil. A more careful minimization and pre-production run equilibration scheme, where the newly-built regions of the structure are allowed to relax while keeping the remainder of the model fixed or heavily constrained, may solve the observed issues, but is perhaps beyond the scope of this study. However, the possibility should be mentioned.

Furthermore, looking at the data, the authors show an increase in radial standard deviation, but looking at the supplemental data, the actual value of the tropomyosin radius in the middle part of the model is in good agreement with the x-ray data (right about 3.95 nm). What this suggests is that the tropomyosin is merely moving in a wider band centered around the same mean. Which does not mean that tropomyosin is losing key contacts and flying away during the whole simulation.

In Table 2, the authors report the persistence length of actin in their simulations as around 8 nm. This seems really low to me, I believe the measured persistence length of actin by experiment is around 18 micrometers. Is that a typo in the table? Otherwise it suggests that the actin in the simulations is considerably more floppy than in the cell.

That observation then brings me to my last point, if the tropomyosin radius from the actin axis is increased, could that just be a result of the underlying actin moving away from the actin axis as well? The persistence length measurement suggests that the next actin on the long pitch helix (only 5.5 nm away) is far off a straight line. Therefore, the authors need to show data from the simulations that are independent of the alignment protocol or the actin dynamics to concretely determine if tropomyosin is truly flying off actin. These would be measurements of key actin-tropomyosin residue contact distances or times, for instance. Another possibility is to look at a more global measure such as interaction energies (Coulombic or van der Waal’s) between tropomyosin and actin to show a significant reduction in tropomyosin-actin interactions.

I also found the graphs to be very busy and the x-axis is hard to relate to tropomyosin. Most readers would be interested in the residue numbers and will have a hard time figuring out where in the tropomyosin sequence they are by knowing the distance in nm from a center point.

Author Response

We are very grateful for the useful and helpful comments, questions, and suggestions. We revised the MS according to the recommendation. Our point-to-point responses to the comments and questions are listed below.

However, the data used to support the conclusions and the proposed source of the errors is incomplete. For instance, a simple explanation for the poor behavior of the tropomyosin model in the simulations would be the extension of the tropomyosin the authors used with their homemade programs to fill out missing parts of the model before simulation.

The homemade software was used only for improving the quality of the Tpm structure between residues 20 and 70 in the N-terminal part and residues 189-280 in the C-terminal part in Model 1, the description of the procedure is added in the Methods section of the revised MS. The missing parts of Tn-T were filled in using available software AlphaFold and Modeller for both models. A significant disorder of pseudo-repeat 2 of a Tpm strand next to the Tpm overlap junction was observed for Model 2, which used the 7utl structure recently deposited to PDB, not for Model 1, a part of which was built with homemade software. The difference between Models 1 and 2 and their behavior during MD simulations is explained in the revised MD.

Another potential source of error is the TnT midpiece structure the authors built. Neither of these were independently tested to see what their effect would be on the global structure. The effects on radial fluctuation reported could be a result of poor contacts in the initial structure from these additional sections which then propagate out into weak points in the tropomyosin coiled coil.

We performed a 204.8 ns long MD simulation of the F-actin-Tpm complex without Tn (and, respectively, without the Tn-T insertions) for Model 2. The radial and angular fluctuations of the Tpm strand in the overlap zone still occurred, although their amplitude decreased while Tpm-actin contacts improved with time during the long MD trajectory. A figure with these results was added as a part of the Supporting Material of the revised MS. This observation suggests that the inter-strand links were probably not the reason for the local Tpm destabilization. We added an explanation in the Discussion section of the MS.

The contact between Tpm and actin monomers estimated by the lifetime of hydrogen bonds between Lys326 and Lys328 actin residues and negatively charged Tpm residues was preserved during MD trajectories except for edges of the model structures and an actin monomer and one of four Tpm molecules in Model 2. A figure illustrating this result was added to the revised MS.

A more careful minimization and pre-production run equilibration scheme, where the newly-built regions of the structure are allowed to relax while keeping the remainder of the model fixed or heavily constrained, may solve the observed issues, but is perhaps beyond the scope of this study. However, the possibility should be mentioned.

We are grateful for the advice. This is indeed beyond the scope of this study. However, we will certainly follow it and do such an analysis. A possibility of local minimization of the new-built Tn-T loops before MD simulation of the whole multiprotein structure is mentioned in the revised MS.

Furthermore, looking at the data, the authors show an increase in radial standard deviation, but looking at the supplemental data, the actual value of the tropomyosin radius in the middle part of the model is in good agreement with the x-ray data (right about 3.95 nm). What this suggests is that the tropomyosin is merely moving in a wider band centered around the same mean. Which does not mean that tropomyosin is losing key contacts and flying away during the whole simulation.

Tpm indeed never flew away from F-actin during our MD simulation. Except for very end regions, Tpm remained at a position on F-actin that agrees with various experimental data. However, local fluctuations near the Tpm overlap zone for Model 2 were high enough for breaking h-bonds and the local opening of a myosin-binding site on actin and compromising the blocked state of the whole regulatory complex.

In Table 2, the authors report the persistence length of actin in their simulations as around 8 nm. This seems really low to me, I believe the measured persistence length of actin by experiment is around 18 micrometers. Is that a typo in the table? Otherwise it suggests that the actin in the simulations is considerably more floppy than in the cell.

We are very sorry for the typo (nm instead of µm) that probably misled the reviewer. Our estimate of the persistence length, ξ, was at the low edge of the range determined in various experiments: ξ = 8-9 µm [Isambert et al., 1995]; ξ = 16.7–19 µm [Ott et al., 1993; Gittes et al., 1993], ξ = 8.5 µm [Nabiev et al., 2015]. In the text of the MS we refer to Deriu et al (2011) paper where experimental values are carefully discussed.

That observation then brings me to my last point, if the tropomyosin radius from the actin axis is increased, could that just be a result of the underlying actin moving away from the actin axis as well? The persistence length measurement suggests that the next actin on the long pitch helix (only 5.5 nm away) is far off a straight line. Therefore, the authors need to show data from the simulations that are independent of the alignment protocol or the actin dynamics to concretely determine if tropomyosin is truly flying off actin.

We suspect that the point mainly results from the typo. We are deeply sorry for that. Tropomyosin did not fly off actin except at the ends of the structure. That was Tn-C that remained bound to Tn core but flew away from Tpm.

These would be measurements of key actin-tropomyosin residue contact distances or times, for instance. Another possibility is to look at a more global measure such as interaction energies (Coulombic or van der Waal’s) between tropomyosin and actin to show a significant reduction in tropomyosin-actin interactions.

We are grateful to the reviewer for the suggestion to analyze Tpm-actin contacts during the MD trajectories. We determined the lifetime of hydrogen bonds (h-bond) between actin residues Lys326 and Ly328 with conserved Tpm negatively charged residues during all three MD trajectories – the parameter that indicate strong interaction. The results are shown in new Fig. 5 of the revised MS. All Tpm pseudo-repeats except those on the ends of the structure and one in the middle on strand 1 had h-bonds with neighbor actin monomers throughout most of the MD trajectory.

I also found the graphs to be very busy and the x-axis is hard to relate to tropomyosin. Most readers would be interested in the residue numbers and will have a hard time figuring out where in the tropomyosin sequence they are by knowing the distance in nm from a center point.

We have redrawn the graphs in the MS and added graphs with the residue numbers to Supporting Materials as suggested.

Reviewer 3 Report

In the manuscript 'The Molecular dynamics assessment of mechanical proerties of the thin filaments in cardiac muscle' submitted by Koubassova et al. to the Int J Mol Sciences, the authors used in silico modeling for the cardiac actin filaments. 

I think the modeling is really interesting. However, I think the manuscript can be improved by explaining that mutations in the different sarcomere proteins cause severe cardiomyopathies. For example you should reference the follwing interesting book chapter explaining the genetic site of this interesting topic:

Gerull, Brenda, Sabine Klaassen, and Andreas Brodehl. "The genetic landscape of cardiomyopathies." Genetic Causes of Cardiac Disease (2019): 45-91.

Author Response

We are very grateful for the evaluation of our work and helpful suggestion.

However, I think the manuscript can be improved by explaining that mutations in the different sarcomere proteins cause severe cardiomyopathies. For example you should reference the follwing interesting book chapter explaining the genetic site of this interesting topic: Gerull, Brenda, Sabine Klaassen, and Andreas Brodehl. "The genetic landscape of cardiomyopathies." Genetic Causes of Cardiac Disease (2019): 45-91.

We expanded the discussion of the use of structural models for understanding the mechanisms of the genetic cardiomyopathies caused by mutations in the thin filament proteins and added more references including one suggested by the reviewer.

Round 2

Reviewer 2 Report

The authors have addressed all concerns I raised previously, and even made some additions that further enhance the paper. 

There is one typo on page 6, paragraph 1, in the phrase "especially its second half" there should be no apostrophe in the word "its".